# Sappanone A Prevents Left Ventricular Dysfunction in a Rat Myocardial Ischemia Reperfusion Injury Model

**DOI:** 10.3390/ijms21186935

**Published:** 2020-09-21

**Authors:** Woori Jo, Byung Sun Min, Hee-Young Yang, Na-Hye Park, Kyung-Ku Kang, Sijoon Lee, Sehyun Chae, Eun Sook Ma, Woo-Chan Son

**Affiliations:** 1Laboratory Animal Center, Daegu-Gyeongbuk Medical Innovation Foundation, Daegu 41061, Korea; c2dar@dgmif.re.kr (W.J.); yhyang@dgmif.re.kr (H.-Y.Y.); pnh0211@dgmif.re.kr (N.-H.P.); kangkk@dgmif.re.kr (K.-K.K.); sjlee1013@dgmif.re.kr (S.L.); 2Department of Medical Science, Asan Medical Institute of Convergence Science and Technology, Asan Medical Center, University of Ulsan College of Medicine, Seoul 05505, Korea; 3College of Pharmacy, Catholic University of Daegu, Gyeongsan 38430, Korea; bsmin@cu.ac.kr; 4Korea Brain Bank, Korean Brain Research Institute, Daegu 41062, Korea

**Keywords:** acute myocardial infarction, coronary artery ligation, myocardial ischemia/reperfusion injury, diastolic function, sappanone A

## Abstract

The incidence of myocardial infarction, among the causes of cardiovascular morbidity and mortality, is increasing globally. In this study, left ventricular (LV) dysfunction, including LV systolic and diastolic function, was investigated in a rat myocardial ischemia/reperfusion injury model with echocardiography. The homoisoflavanone sappanone A is known for its anti-inflammatory effects. Using echocardiography, we found that sappanone A administration significantly improved LV systolic and diastolic function in a rat myocardial ischemia/reperfusion injury model, especially in the early phase development of myocardial infarction. Based on myocardial infarct size, serum cardiac marker assay, and histopathological evaluation, sappanone A showed higher efficacy at the doses used in our experiments than curcumin and was evaluated for its potential to improve LV function.

## 1. Introduction

Acute myocardial infarction (AMI) has recently emerged as one of the leading causes of increased cardiovascular morbidity and mortality worldwide [1]. In the context of myocardial ischemia, systolic myocardial dysfunction usually occurs before electrocardiogram (ECG) changes or chest pain occur [2], so it is important to specifically evaluate left ventricular (LV) when diagnosing and monitoring cardiovascular disease. Echocardiography, the best prognostic indicator for MI patients, is a well-established and non-invasive diagnostic tool for evaluating LV function [3]. LV diastolic function has become more important in recent years. Additionally, MI patients with LV diastolic dysfunction have poorer postoperative outcomes, such as heart failure or cardiac death, than those with only LV systolic dysfunction [4,5].

In cardiovascular research, rat myocardial ischemia/reperfusion (I/R) injury models are the most commonly used animal models, and this animal model has been commonly used in efficacy studies of new drugs, stem cell therapy, and mechanism studies [6,7,8,9]. With scientific and technological advances, precise and accurate echocardiographic evaluation is possible even with rodents that have a mean heart rate of over 300 bpm [10]. With modern technology, it is possible to set a rat myocardial I/R injury model induced by transient ligation of left anterior descending (LAD) and proceed with echocardiographic evaluation of LV function, including systolic and diastolic dysfunction [11,12].

Homoisoflavanones are a small class of natural products, which include sappanone A, a compound isolated from the heartwood *Caesalpinia sappan* that has proven inhibitory activity against nitric oxide (NO) production as well as against the expression of inducible nitric oxide synthase (iNOS) and cyclooxygenase-2 (COX-2) in lipopolysaccharide (LPS)-stimulated RAW264.7 cells [13]. This compound induces heme oxygenase (HO)-1 protein and increases nuclear translocation of the nuclear factor-E2-related factor 2 (Nrf2) as well as the expression of Nrf2 target genes, such as NAD(p)H:quinone oxidoreductase 1 (NQO1), in RAW264.7 cells [14]. Additionally, sappanone A has been shown to inhibit RANKL-induced osteoclastogenesis in mouse bone marrow macrophages (BMMs) and suppress inflammation-induced bone loss in a mouse model [15].

Despite the various anti-inflammatory effects of sappanone A, there has been no evaluation of its cardiovascular protective effects in vivo, so we proceeded with this study to show the effectiveness of sappanone A on LV dysfunction using echocardiography in a rat myocardial I/R injury model. To evaluate the effectiveness of sappanone A in a rat myocardial I/R injury model, we assessed LV systolic and diastolic function via echocardiography, performed serum cardiac marker assays, histopathological examination, and mechanism analysis, and we determined LV infarction size. Our findings will help future MI patients in the early stages of cardiovascular disease development.

## 2. Results

### 2.1. Experimental Design, Gross Examination, Myocardial Infarct Size, and Serum Cardiac Marker Results

To evaluate the cardioprotective effects of sappanone A isolated from the heartwood *C. sappan*, as previously described [13] (Figure 1A), transient LAD ligation surgery and administration proceeded according to the experimental design (Figure 1B). Figure 1C shows the representative gross picture of the MI vehicle group, and the infarct regions were found below the ligation site at the apex area (yellow arrow). Also, Figure 1D shows the representative LV slices of the treatment groups for evaluating LV myocardial infarct size. The infarct size of the curcumin- and sappanone A-treated groups decreased compared with the vehicle-treated MI group, and tended to alleviate myocardial infarction, but only the sappanone A-treated group had statistical significance. The area of each LV tissue slice was divided into four areas: PM, proximal medial, distal medial, and apex areas. The mean and SEM values of the infarct size (%) for each slice are shown in Table 1. In particular, Sappanone A treatment was associated with significantly smaller infarct sizes in the distal medial and apical areas.

Serum Creatine Kinase MB (CK-MB), Lactate dehydrogenase (LDH), and Aspartate aminotransferase (AST) are important variables for the assessment of MI-induced models due to myocardial cell damage and rupture [16]. As the results of the cardiac marker assays, serum CK-MB, LDH, and AST levels in the rat myocardial I/R injury model group (MI+Veh) were significantly increased compared with the normal control group. The curcumin and sappanone A treatment groups tended to have lower levels of CK-MB, LDH, and AST, and sappanone A significantly reduced the LDH and AST levels compared with those of the MI+Veh group out of the MI groups (Figure 1E).

### 2.2. Echocardiographic Results

Overall, cardiac function did not recover from day 1 to day 4, including systolic and diastolic function (Figure 2). EF and FS are representative values for evaluating the LV systolic function, and E′ and E/E′ ratio are the representative values for evaluating the LV diastolic function. The LV systolic function tends to increase in the sappanone A 50 mg/kg dose group compared with the MI + vehicle group. LV diastolic dysfunction did not recover in the MI + vehicle group until day 4, whereas in the curcumin-treated and sappanone A-treated groups, the absolute value of the E/E′ ratio was significantly decreased compared with the MI + vehicle group, which confirmed the cardioprotective effect on LV diastolic dysfunction. Also, sappanone A showed a higher efficacy at this dose than curcumin. In particular, the LV diastolic dysfunction was significantly alleviated in association with curcumin and sappanone A, and the E/E′ ratio in the sappanone A group improved the LV diastolic function to almost normal levels. The detailed results are described in Table 2.

### 2.3. Histopathological Results

Representative photomicrographs of tissue stained with hematoxylin and eosin (H&E) are shown in Figure 3A,B and Masson’s trichrome are shown in Figure 3C. In H&E-stained slide sections, inflammatory cell infiltration is shown in epicardial, myocardial, and endocardial areas. Masson’s trichrome staining revealed areas of marked fibroblast and collagen deposition. Figure 3A shows histopathological changes in papillary muscle (PM), medial, and apex areas. Representative photomicrographs of interest with high magnification are shown in Figure 3B (×40, ×400, ×1500). Mixed cell (mixed state of lymphocytes and monocytes), lymphocyte (arrows) and mononuclear cells (arrow heads) are shown in myocardial region (Top, ×400) and mononuclear cells (arrow heads) with purulent (asterisk) are shown in endocardial region (Bottom, ×400). Representative cell image of the mature lymphocyte (arrows) and mononuclear cells (arrow heads) are shown in Figure 3B with high magnification (×1500). Sappanone A was associated with a significant reduction in inflammatory cells infiltration in all epicardial regions and the medial myocardial region. The degree of fibrosis in all regions in the MI + Veh group was higher than that seen in curcumin-treated and sappanone A-treated groups, although these differences were only statistically significant in PM area (Figure 3C).

### 2.4. Sappanone A Treatment Results in Changes of Multiple Cellular Processes in the Rat Myocardial I/R Injury Model

To investigate molecular signatures affected by sappanone A in a rat myocardial I/R injury model, we performed mRNA sequencing of left ventricle tissues from the normal group, MI group, MI + Curcumin group and MI + Sappanone A group, and we compared mRNA abundances between the different groups (Figure 4A). A total of 3568 DEGs with false discovery rates < 0.05 and fold changes >1.5 were identified (Materials and Methods) from the MI versus normal, MI + Curcumin versus MI, and MI + Sappanone A versus MI (2163, 2020, and 793 DEGs, respectively) comparisons (Figure 4B and Appendix A). Large numbers of DEGs were identified from the MI + Sappanone A versus MI (2020 DEGs) comparison, whereas only 793 DEGs were identified from the MI + Curcumin versus MI comparison. Also, among these DEGs, 883 and 392 genes were shared with the DEGs identified from the MI versus normal comparison (40.8% and 18.1% of the 2163 DEGs, respectively) (Figure 4B). These data indicate that gene expression can be significantly altered by the sappanone A in the rat myocardial I/R injury model. To systematically investigate the cellular processes associated with sappanone A in this rat MI model, these shared DEGs were categorized into six clusters (C1-6) based on their differential expression patterns (Figure 4C).

The cellular processes represented by the genes in the individual clusters were identified by performing enrichment analyses of GOBPs using DAVID software [17]. Among the six clusters, C1/3 and C4/6 showed up- and down-regulation in their abundances by MI, respectively, but sappanone A treatment inhibited these alterations. The GOBP enrichment analysis revealed that the genes in C1/C3 were mainly involved in immune and inflammatory responses, cytokine production, neutrophil migration, NF-kB signaling, and apoptosis (Figure 4D), while C4/6 were involved in glucose and fatty acid metabolism, mitochondrial organization, cellular oxidant detoxification, reactive oxygen species metabolism, and cardiac muscle contraction (Figure 4E). Particularly, the genes involved in inflammatory responses were strongly restored in terms of their expression by sappanone A (*p* < 10^−4^) (Figure 4F). The results showed that sappanone A affects a broad range of cellular processes. 

### 2.5. Inhibitory Effects of Sappanone A on Myocardial Infarction-Related Processes

To examine the collective actions of signaling pathways, a network model describing interactions among the DEGs was reconstructed. The network model showed that sappanone A treatment down-regulated pro-inflammatory pathways activated by MI (*Tlr*, *Tgfb*, *TNF*, and *Ifng* signaling pathways) as well as the complement cascade, which increases inflammation and apoptosis pathway (Figure 5A). 

Finally, we confirmed downregulation of the representative genes involved in inflammatory responses (*Tgfb1*, *Tgfb2*, *Cd4*, *Cd8a*, *Il18*, *Pik3cd,* and *Tnfrsf1a*) and apoptosis (*Casp3*) in the MI + Sappanone A group, compared with in the MI groups (Figure 5B and Appendix A). Taken together, these results suggest that sappanone A attenuate MI-related processes including inflammation in the rat myocardial I/R injury model.

## 3. Discussion

In this in vivo model, the positive effect of sappanone A on the early stage of MI was confirmed by assessing the infarct size and serum cardiac marker (CK-MB, LDH, and AST) on day 1 after MI surgery and the LV systolic and diastolic function using echocardiography on day 1 and 4. In addition, histopathological changes (H&E and Masson Trichrome) and mRNA sequencing were analyzed to study potential changes in cellular pathways on day 4 after MI surgery. These data demonstrate that sappanone A successfully attenuated LV dysfunction induced by LAD ligation in a rat myocardial I/R injury model.

Acute MI remains one of the leading causes of increased cardiovascular morbidity and mortality worldwide [1,18], and LV diastolic dysfunction during acute MI is among the important indicators of poor surgical outcomes and recurrences [4]. Echocardiography is useful for detecting LV dysfunction and is the best prognostic indicator for human MI patients [3,19].

Rat myocardial I/R injury models are useful, and we especially focused on LV diastolic function and systolic function using echocardiography. In the present study, the rat myocardial I/R injury model was successfully induced by LAD ligation and reperfusion using a snaring technique and area at risk was normalized by TTC/EB double staining. Curcumin and sappanone A were daily administered before ischemia from on the day of surgery to four days after surgery for five days, which was a relatively short exposure period compared with previous studies [8,11,12], to investigate their early effects on MI. We continuously emphasized the importance of the LV diastolic and systolic dysfunction to better simulate the clinical cardiac function assessment of MI patients. We found that the rats with MI exhibited decreased ejection fraction and fractional shortening values, reflecting LV systolic dysfunction; they also had reduced E′ values and increased E/E′ values, reflecting LV diastolic dysfunction, and this tends to be the same in human MI patients. Patients with acute coronary syndrome usually have increased end-diastolic filling pressure of the left ventricle, leading to early aortic valve closure. If stroke volume is decreased sequentially, the incoming blood flow in late systole decreases [20], which causes disturbances of LV diastolic function, leading to large infarctions [21]. Since elevation of LV filling pressure is the key indicator of poor outcomes in humans [22,23], the E/E′ ratio can be used to evaluate LV diastolic dysfunction, given its tendency for independence from LV systolic function, heart rhythm abnormalities, and LV hypertrophy [24]. Therefore, the evaluation of LV diastolic function is important for MI patients, and our echocardiographic method is a good indicator in terms of prognostic prediction and efficacy evaluation in the context of new drug development or heart disease.

Curcumin is the natural yellow pigment extracted from the rhizomes of the plant *Curcuma longa,* and its cardioprotective effect in a rat chronic MI model induced via the TGFß/Smad-mediated signaling pathway has been studied [8]. An acute myocardial I/R injury rat model has also been used to study the protective effect of curcumin in association with enhanced STAT3 phosphorylation [25]. Recently, the effects of curcumin nanoparticles in isoproterenol-induced MI have been demonstrated [26]; however, the age and the bodyweight of rats, the ischemic period of LAD ligation, and the dosage and total treatment period were different from our rat myocardial I/R injury model protocol. Using curcumin as a positive control group, our new study intended to identify the effects of sappanone A on acute MI.

In a previous experiment, we found no differences in the serum chemistry values for cardiac markers in normal and surgically-induced MI model rats on the seventh day after surgery, so the sampling was performed on the first day after transient LAD ligation surgery, which was earlier than previous experiments; this was done to consider the time-bound effect of the enzymes in serum, and then statistically significant changes were detected when MI occurred. After MI was induced, myocardial cells were damaged and ruptured, cardiac enzymes were released into the blood, and a significant decline in mean serum AST was observed in the sappanone A-treated rats, relative to the MI+Veh rats. In addition, the mean myocardial infarct size of the sappanone A-treated rats was significantly smaller in the distal medial and apex regions compared with that of the MI+Veh group. Additionally, the histopathological findings of myocardial tissue from rats in the MI group were clearly different from those of rats in the normal group. The pathogenic consequences of MI are usually seen in the main coronary arteries and myocardium [27]. In this study, based on the histopathological examination of left ventricles on day 4 after MI surgery, we could determine that it is a condition that progresses from acute or subacute to sub-chronic stage. Advanced lesions were rarely observed and the inflammatory cells of the lymphocyte and mononuclear cell lineages, which are mainly seen in the sub- chronic inflammatory status, are distributed more significantly than neutrophil or eosinophil, which is mainly seen in the acute stage. In addition, the microvascular obstruction and neutrophils in H&E staining were rarely seen on the day 4 after MI surgery. To evaluate the effect of Sapponone A on microvascular obstruction and neutrophil infiltration, it could be better to conduct histopathologic examination in earlier time such as day 1 after MI surgery. In H&E-stained slide sections of each transverse cardiac region, inflammatory cells infiltration including lymphocytes and mononuclear cells, and purulent lesions were examined by grading, and sappanone A significantly reduced inflammatory cells infiltration in all epicardial regions and the medial myocardial region with the consistency of peer-review. In addition, it seems sappanone A influence more on the lymphocyte based on our histopathologic results in heart samples, but without the results of histopathological examination of other organs such as spleen, bone marrow, etc., there are still limits to confirming the greater effect of sappanone A on lymphocyte based on heart results alone. Thus, the effects of sappanone A on each cell type of inflammation and its direct/indirect relationship to MI will be discussed in the further studies. Fibrosis area percentages in Masson’s trichrome–stained slide sections were only statistically different among the groups in PM area.

The echocardiographic results confirmed the successful generation of our acute rat myocardial I/R injury model. The echocardiographic data showed that LV cardiac dysfunction did not recover during the early phase of ischemic-reperfusion injury in this rat model throughout the experimental period. However, sappanone A had cardioprotective effects on acute myocardial ischemia according to the significantly improved LV systolic and diastolic function and reduction in ischemic lesions. Also, there are some limitations to using E′ and E/E′, which reflect only the global LV function. However, further studies using quantitative evaluation with strain speckle tracking echocardiography are planned for evaluating regional LV function and filling dynamics [28].

The alteration of molecular signatures by sappanone A had not previously been systematically explored using a rat myocardial I/R injury model. In this study, gene expression profiling was used to identify molecular signatures affected by sappanone A in a rat myocardial I/R injury model. Sappanone A treatment altered the mRNA expression level of 2020 genes involved in various cellular processes, including 66 genes involved in inflammatory responses (Figure 4F and Appendix A). These genes are likely to be involved in myocardial infarction-related pathological features. For example, *Tgfb1* and *Tgfb2* mRNA abundances were decreased by sappanone A. The *Tgfb* family critically regulates the inflammatory response, angiogenesis, and fibrosis under myocardial infarcts [29]. Furthermore, the network model suggested *Tlr, Nfkb, Tnf, Ifng* signaling pathways associated with disease pathogenesis [30]. Although the sappanone A showed potent effects in decreasing the expression of pro-inflammatory factors, the exact mechanisms behind the anti-inflammatory effects of sappanone A and improved heart function are not clearly understood. The reduced pro-inflammatory factors predominantly by sappanone A may act as a potential link of the inflammation to the alleviation of the LV diastolic and systolic dysfunction [31,32,33,34,35]. In this study, we focused on the effects of sappanone A on myocardial infarction and the restoration of the MI-perturbed gene expression profiles (e.g., inflammation-related pathways). It can be considered valuable as an initial comparative study, and further mechanistic studies are warranted to elucidate the functional link of the attenuated inflammation to the improved myocardial infarction phenotypes. 

To the best of our knowledge, ours was the first study to demonstrate the promising positive effects of sappanone A on LV dysfunction in a rat myocardial I/R injury model using echocardiography. The rat myocardial I/R injury model is a good representation of human acute MI, allowing the cardioprotective effects of sappanone A to be evaluated. These data contribute to the understanding of the effects of sappanone A on the development of AMI and provide a clear rationale for the use of sappanone A in high-risk patients.

## 4. Materials and Methods

### 4.1. Chemicals and Reagents

Sappanone A was obtained from Eun Sook Ma, Daegu Catholic University, Gyeongsan, Korea (Figure 1A), and extracted using methods previously described [13,14].

### 4.2. Animals, Husbandry, and Experimental Design

Sprague Dawley rats (8-week-old adult males, mean weight: 285.33 ± 5.09 g) were purchased from Koatech (Kyungki, Korea). The study design was approved by the Institutional Animal Care and Use Committee of Daegu-Gyeongbuk Medical Innovation Foundation (3 January 2019, DGMIF-18041705-01). The rats were housed in environmental conditions with a temperature of 22 ± 1 °C, a relative humidity of 50% ± 10%, 12 h light/dark cycles, illumination at 150–300 Lux, and ventilation 10–20 times/h. The rats were monitored every hour for 24 h and maintained within an acceptable environment throughout the study. There were three rats per cage at the beginning of the study, and they were fed an autoclaved pellet diet (SAFE + 40RMM; SAFE Diets, Augy, France) *ad libitum.*

The experimental design is shown in Figure 1B. The rats were divided into four groups (eight rats/group) as follows: (A) Normal group, (B) MI group, (C) MI + Curcumin 25 mg/kg group, (D) MI + Sappanone A 50 mg/kg group. The rats in groups A and B received vehicle (10% dimethyl sulfoxide, DMSO, cat.co. 472301, Sigma-Aldrich, St. Louis, MO, USA and 90% polyethylene glycol, PEG400, cat.co. 91893, Sigma, St. Louis, MO, USA), and the rats in groups C and D were treated with 25 mg/kg curcumin, and 50 mg/kg sappanone A dissolved in vehicle by oral injection daily for 5 days (before ischemia from the day of surgery to 4 days after surgery).

### 4.3. Induction of Myocardial I/R Injury

Animals were anesthetized with alfaxalone (50 mg/kg, IP) and xylazine (5 mg/kg, IP). After anesthetization, the rats were intubated and ventilated using a respirator (Harvard Apparatus VentElite, Holliston, MA, USA) and maintained on a tidal volume of 3.0 mL/kg and a respiratory rate of 60 breaths/min. Intraoperatively, the rats were placed on a heated plate and monitored by ECG, and the rat myocardial I/R injury model involved ligation of the left anterior descending (LAD) coronary artery for 30 min, as described previously [11,12]. MI was confirmed by the paleness of the apical region of the left ventricle and S–T segment elevation on the ECG [11,12]. 

### 4.4. Echocardiographic Analysis

Echocardiography was performed on days 1 and 4 after the induction of MI, using Vevo2100 (FUJIFILM VisualSonics, Inc., Toronto, ON, Canada). The rats were anesthetized with alfaxalone (50 mg/kg, IP) and xylazine (5 mg/kg, IP) and were monitored by ECG while in a supine position and maintained at a body temperature of 37 °C.

Echocardiographic parameters were in accordance with the American Society of Echocardiography guidelines [1]. The images of the LV parasternal short-axis (SAX) view at the papillary muscle level and the apical four-chamber view were obtained to evaluate LV systolic and (especially) diastolic function using B-mode, M-mode, Doppler color flow, pulsed wave Doppler, and tissue Doppler, as described previously [11,12].

### 4.5. Myocardial Infarct Size

On day 1 after MI surgery, the rats (5 rats per group) were euthanized with isoflurane, and their hearts were excised. After reperfusion with 0.9% normal saline, the hearts were perfused with 2 mL of 2 % Evans blue (Sigma, St. Louis, MO, USA) from the aorta. Then, the hearts were sectioned into 2 mm transverse slices and immersed in a 1% solution of 2,3,5-triphenyltetrazolium chloride (TTC, Sigma, St. Louis, MO, USA) at 37°C for 15 min in the dark. The infarct area, area at risk, and left ventricle area were analyzed using ImageJ software (National Institutes of Health, Bethesda, MA, USA) and expressed as a ratio of the area of the ischemic zone over the LV area (IA/LV). 

### 4.6. Serum Chemistry of CK-MB, LDH, and AST 

On day 1 after MI surgery, the rats were anesthetized, and the abdominal cavity was opened. The blood was collected in serum separate tubes (SST tube, cat.367989, BD Inc., Franklin Lakes, NJ, USA), centrifuged at 3000 rpm for 10 min, and the serum was separated for use in cardiac marker assays. The serum creatine kinase-MB (CK-MB), lactate dehydrogenase (LDH), and aspartate aminotransferase (AST) were measured with a TBA-120FR automated chemistry analyzer (Toshiba, Tokyo, Japan).

### 4.7. Histopathological Analysis

On day 4 after MI surgery, rat hearts were harvested after removal of blood for histological assessment (*n* = 3) and fixed in 10% neutral buffered formalin (BBC Biochemicals, Mount Vernon, WA, USA). The formalin-fixed heart tissues prepared for analysis using a tissue processor (Thermo Fisher Scientific, Inc., Runcorn, UK). The paraffin-embedded tissue blocks were cut at a 4 μm thickness, mounted onto glass slides, and stained in hematoxylin (YD-Diagnostics, Kyungki, Korea) and eosin (BBC Biochemicals, Mount Vernon, WA, USA) using an autostainer (Dako CoverStainer; Agilent, Santa Clara, CA, USA). To evaluate fibrosis, tissue sections were stained using a Masson’s trichrome staining kit according to the manufacturer’s instructions (ScyTek Laboratories, West Logan, UT, USA). After staining, the parasternal short-axis area with papillary muscle (PM), and the medial and apex areas of the left ventricle were scanned with a slide scanner (Pannoramic SCAN II; 3DHISTECH, Budapest, Hungary) and captured using a slide viewer (Case Viewer; 3DHISTECH). In light microscope, especially H&E slides, the distinction between mature lymphocytes and other three types of cells (immature lymphocyte, mature/immature monocyte) was possible to score, and shown in Figure 3 and Appendix A by three independent pathologists. However, distinction among these three types of cells (immature lymphocyte, mature/immature monocyte) are almost impossible under the microscope with a H&E slide. Therefore, we collectively called these three type of cells as mononuclear cells. Based on the previous explanation, the term “mixed cells” means the mixed state of lymphocytes and monocytes as shown in Appendix A. ImageJ program (provided by the NIH) was used to perform the morphometric analysis of fibrosis. In Masson’s trichrome-stained sections, the blue area (collagen fiber) was measured in comparison with the total red area (left ventricle) and the results were shown in Figure 3C.

### 4.8. mRNA Sequencing and Data Analysis

For gene expression profiling, total RNAs were obtained from the LV tissues including inter ventricular septum of rat hearts from the four groups (normal group, MI group, MI + Curcumin group, and MI + Sappanone A group) using Trizol reagent (Invitrogen Life Technologies, Grand Island, NY, USA) on day 4 after MI surgery. The integrity of the total RNA was analyzed using an Agilent Bioanalyzer. The RNA integrity values for all of the samples were larger than 7. Poly (A) mRNA isolation from total RNA and fragmentation was performed using the Illumina TruSeq Stranded mRNA Sample Prep Kit, according to the manufacturer’s instructions. The adaptor-ligated libraries were sequenced using an Illumina NovaSeq 6000 (Bioneer, Daejeon, Korea). In each condition, the mRNA-sequencing analysis was performed for two biological replicates obtained from independent rats (Appendix A). 

Adapter sequences (TruSeq universal and indexed adapters) were removed using Cutadapt software (version 2.7; https://cutadapt.readthedocs.io/en/stable/), and the remaining read sequences for each sample were aligned to the *Rattus_norvegicus* reference genome (Rnor_6.0) using TopHat2 software (version 2.1.1) with default parameters [36]. After the alignment, the numbers of reads mapped to the gene features (GTF file of Rnor_6.0.90) were counted using HTseq [37]. The read counts for the samples in each condition were then normalized using the TMM (trimmed mean of M-values) normalization function of the edgeR package [38].

### 4.9. Identification of Differentially Expressed Genes (DEGs)

The numbers of reads for the gene features were converted to log_2_-values after adding one (pseudo count) to the read counts. To identify DEGs between four conditions, the previously reported statistical hypothesis test was performed [39]. Briefly, for each gene, a T-statistic value was calculated using Student’s *t*-test in each of the three comparisons (MI group versus normal group, MI + Curcumin group versus MI group, or MI + Sappanone A group versus MI group). For each comparison, the empirical distributions of the T-statistic value for the null hypothesis (i.e., the genes are not differentially expressed) were estimated by performing all possible combinations of random permutations of samples. Using the estimated empirical distributions, adjusted *p* values for Student’s t-test for each gene were calculated. Finally, the DEGs were identified as those that had adjusted *p* values ≤ 0.05 and absolute log2-fold-changes ≥0.58 (1.5-fold). To identify cellular processes represented by the DEGs, the enrichment analysis of Gene Ontology Biological Processes (GOBPs) was performed using DAVID software (https://david.ncifcrf.gov/summary.jsp) [17], and the GOBPs with *p* values < 0.05 were selected as the processes enriched by the DEGs. The network model was reconstructed for the selected DEGs using Cytoscape software (version 3.3.0) [40]. The nodes in the network model were arranged based on the locations and relationships of the corresponding genes in the Kyoto Encyclopedia of Genes and Genomes (KEGG) pathway database [41].

### 4.10. RT-PCR

Total RNA was prepared from the LV tissues of rats with same experimental condition, after which cDNAs were synthesized using a SuperScript™ IV First-Strand Synthesis System for RT-PCR according to the manufacturer’s instructions (Invitrogen Life Technologies, Grand Island, NY, USA). The PCR was conducted by subjecting the samples to the following conditions: initial denaturation at 95 °C for 5 min, followed by 22~27 cycles of amplification by denaturation at 95 °C for 30 s, annealing at 57~59 °C for 30 s, extension at 72 °C for 30 s, and final extension at 72 °C for 5 min. The amplified PCR products were then separated on 1.5% agarose gels and visualized by SYBR Safe staining (Invitrogen Life Technologies, Grand Island, NY, USA). The primer information used was included in Appendix A and the representative bands are shown in Appendix A. Hypoxanthine-guanine phosphoribosyltransferase 1 (HPRT1) was used as a control gene for normalization, and the data were derived from four independent experiments. The densitometric analysis was performed on RNA expression patterns using ImageJ software and the relative value were displayed in Figure 5B. 

### 4.11. Statistical Analysis

Statistical significance was determined using GraphPad Prism 8 (GraphPad Software Inc., San Diego, CA, USA). All data are presented as mean ± standard error of the mean (SEM). For comparisons among multiple groups of one variable (for example, with and without treatments), one-way analysis of variance (ANOVA) with Tukey’s post hoc correction was used (Table 1, Figure 1D and Figure 3). Also, for comparisons among multiple groups of two variables (for example, MI and treatments), two-way ANOVA with Tukey’s post hoc correction was used (Table 2, Figure 1E, Figure 2A–D, and Figure 5B). A *p* value <0.05 was considered statistically significant.

## Figures and Tables

**Figure 1 ijms-21-06935-f001:**
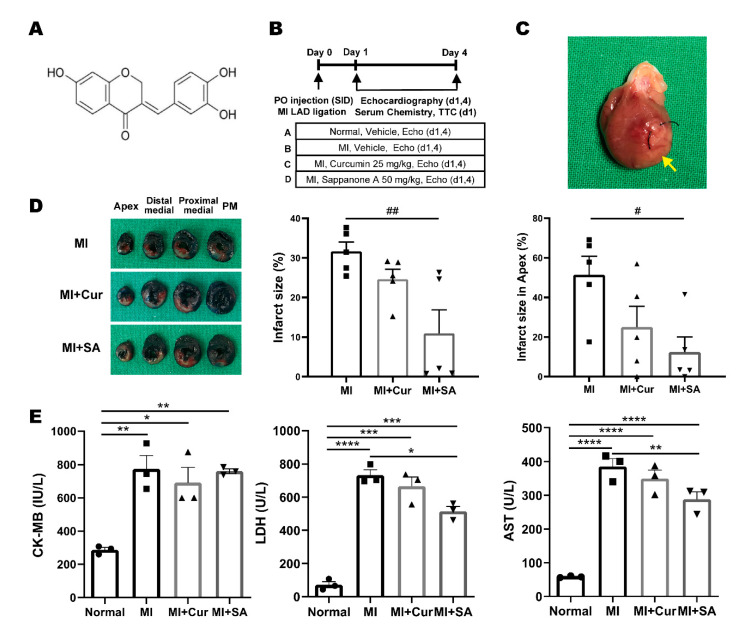
The experimental design and the positive effect on the infarct size and serum cardiac marker. (**A**) Chemical structure of sappanone A. (**B**) The animals were divided into four groups: Group A, Normal control; Group B, MI*+*Vehicle; Group C, MI+Curcumin 25 mg/kg; Group D, MI+Sappanone A 50 mg/kg. (**C**) Representative gross photograph of a heart on day 1 after MI surgery and the infarct area (yellow arrow). (**D**) Representative photographs of left ventricle slices of the groups and the infarct size of the left ventricles of the total and apex areas in each group (*n* = 5/group, right). ^#^
*p* < 0.05 and ^##^
*p* < 0.01 by one-way ANOVA test with Tukey’s post hoc correction. (**E**) Serum chemistry results of creatine kinase MB isozyme (CK-MB, *n* = 3/group), lactate dehydrogenase (LDH, *n* = 3/group), and aspartate aminotransferase (AST, *n* = 3/group) on day 1 after MI surgery. ** *p* < 0.01, *** *p* < 0.001, **** *p* < 0.0001 by two-way ANOVA test with Tukey’s post hoc correction.

**Figure 2 ijms-21-06935-f002:**
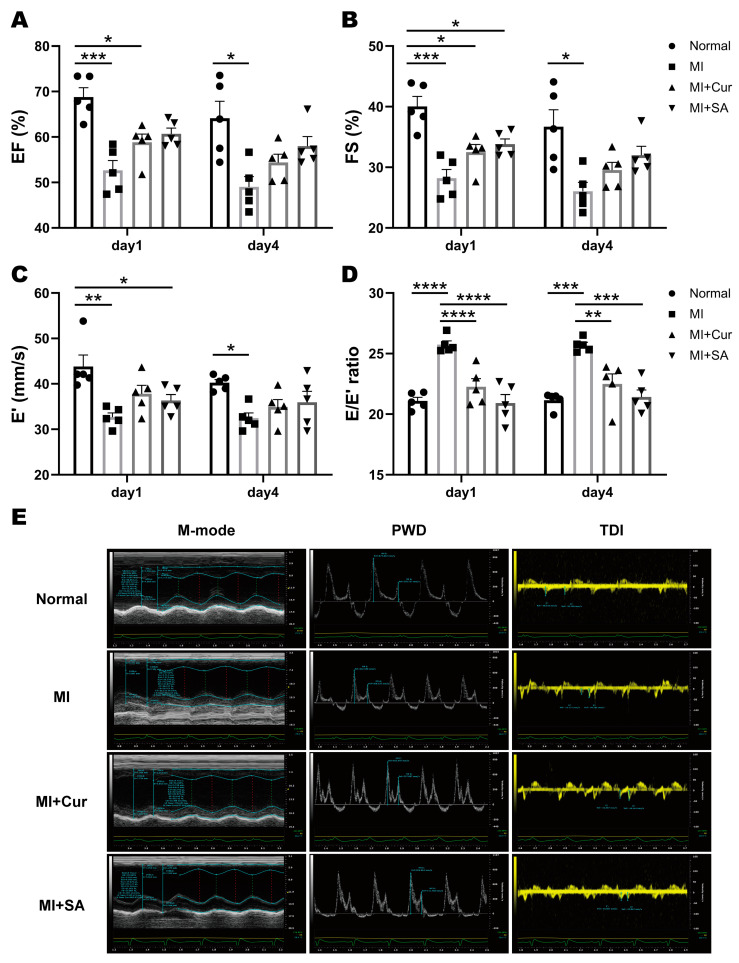
Echocardiographic results (*n* = 5/group) and the representative echocardiographic images of rat left ventricle on day 4 after MI surgery. Sappanone A significantly improved left ventricular (LV) systolic and diastolic function. (**A**) Ejection fraction, EF; (**B**) Fractional shortening, FS; (**C**) Early relaxation velocity on tissue Doppler, E′; (**D**) E/E′ ratio of LV diastolic function; (**E**) Representative echocardiographic imaging on day 4 after MI surgery. * *p* < 0.05, ** *p* < 0.01, *** *p* < 0.001, **** *p* < 0.0001 by two-way ANOVA tests with Tukey’s post hoc correction.

**Figure 3 ijms-21-06935-f003:**
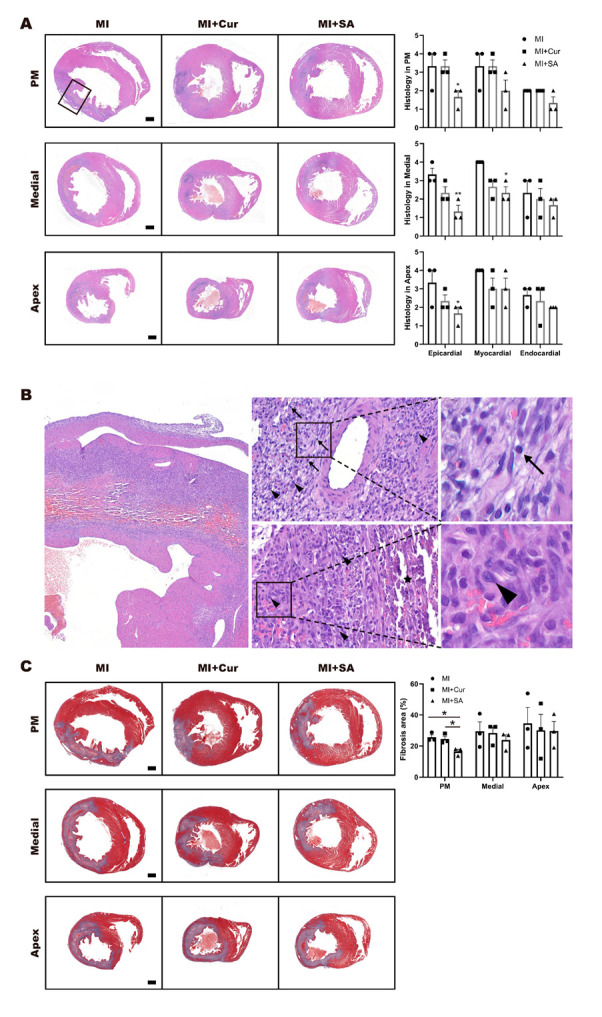
Histopathological images and results in each left ventricle area on day 4 after MI surgery. Representative photomicrographs are shown in hematoxylin and eosin and incidence of histopathological changes (**A**, ×12.5) and representative H&E photomicrographs with high magnification (**B**, ×40, ×400, ×1500). Mature lymphocyte (arrows) and mononuclear cells (arrow heads) are shown in myocardial region (Top, ×400) and mononuclear cells (arrow heads) with purulent (asterisk) are shown in endocardial region (Bottom, ×400). Representative cell image of lymphocyte (arrow) and mononuclear cell (arrow head) can be identified with high magnification (×1500). Representative photomicrographs in Masson’s trichrome and fibrosis area percentage (**C**, ×12.5). * *p* < 0.05, indicate statistically significant differences by one-way ANOVA with Tukey’s post hoc correction. Grading of histopathological changes in each tissue area (papillary muscle, medial, and apex) of the rat left ventricle. Grades 1, 2, 3, and 4 show minimal, slight, moderate, and severe pathological changes, respectively. Values are mean ± standard error of the mean (SEM, *n* = 3).

**Figure 4 ijms-21-06935-f004:**
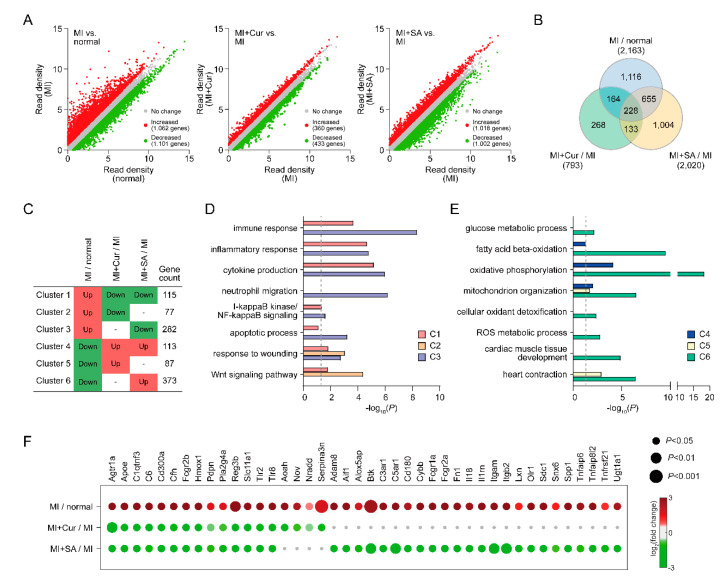
Cellular processes affected by sappanone A. (**A**) Scatter plots of three comparisons (MI versus normal, MI + Curcumin versus MI, and MI + Sappanone A versus MI). (**B**) Relationships among differentially expressed genes (DEGs). (**C**) Clusters (C1-6) of the genes affected by curcumin or sappanone A. Red and green denote up- and down-regulation, respectively. The number of DEGs in each cluster is denoted. (**D**,**E**) Cellular processes represented by DEGs in C1-6. X-axis, −log_10_(P) where P is the enrichment *p*-value calculated in DAVID software. (**F**) DEGs involved in inflammatory responses.

**Figure 5 ijms-21-06935-f005:**
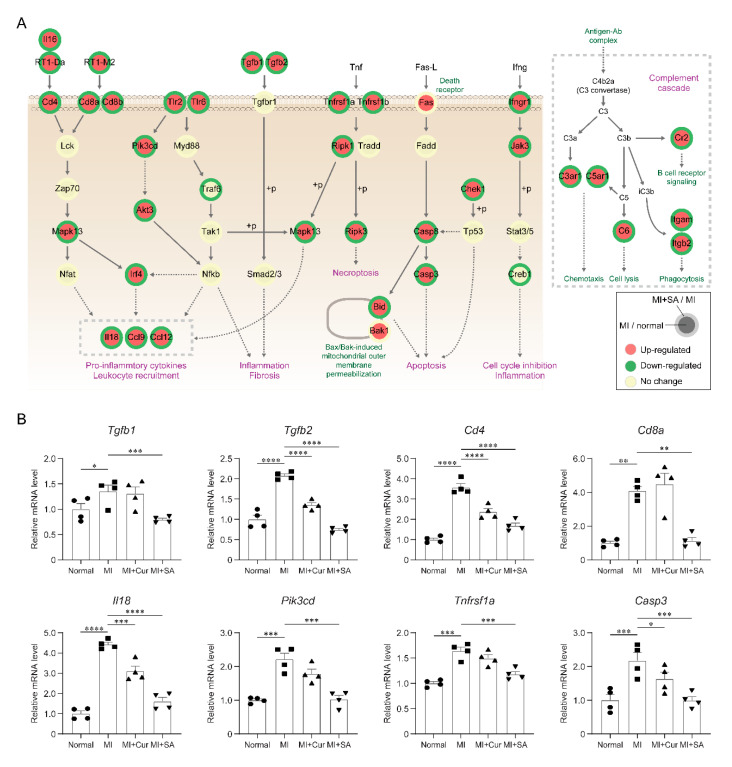
Inhibitory effects of Sappanone A administration on the MI-related processes. (**A**) Network model describing interactions among signaling pathways. Arrows, activation in signaling. “+p”, phosphorylation. (**B**) Confirmation of the predominant downregulation of the eight representative genes involved in the inflammatory responses and apoptosis by RT-PCR. The expression levels were normalized with respect to those in the control group. The normalized data are expressed as the mean ± SEM (*n* = 4 per group). * *p* < 0.05, ** *p* < 0.01, *** *p* < 0.001, **** *p* < 0.0001 by two-way ANOVA tests with Tukey’s post hoc correction.

**Table 1 ijms-21-06935-t001:** Myocardial infarct size (%) in each MI area region measured by TTC staining.

MI Area Region	MI	MI + Cur	MI + SA
Papillary muscle (PM)	20.52 ± 11.54	25.68 ± 7.45	11.53 ± 8.35
Proximal medial	21.52 ± 2.63	23.19 ± 2.38	10.19 ± 6.31
Distal medial	33.09 ± 4.83	24.55 ± 3.12	^#^ 9.64 ± 6.22
Apex	51.46 ± 9.33	^#^ 25.01 ± 10.50	^###^ 12.39 ± 7.64
Total	31.65 ± 2.37	24.61 ± 2.53	^##^ 10.94 ± 5.95

Values are expressed as the mean ± SEM. ^#^ (*p* < 0.05), ^##^ (*p* < 0.01), and ^###^ (*p* < 0.001) indicate statistically significant differences by one-way ANOVA test with Tukey’s post hoc correction.

**Table 2 ijms-21-06935-t002:** Cardiac function measured by echocardiography.

Cardiac Function	Day 1				Day 4			
	Normal	MI	MI+Cur	MI+SA	Normal	MI	MI+Cur	MI+SA
EF, %	68.79 ± 4.59	*** 52.63 ± 4.85	* 58.81 ± 4.12	60.68 ± 2.82	64.13 ± 8.30	* 49.01 ± 5.06	54.39 ± 4.07	57.97 ± 4.70
FS, %	40.05 ± 3.66	*** 28.21 ± 3.18	* 32.48 ± 2.85	* 33.80 ± 1.98	36.69 ± 6.24	* 26.03 ± 3.28	29.55 ± 2.80	31.99 ± 3.30
HR, BPM	256.16 ± 21.28	302.19 ± 48.56	303.10 ± 20.50	259.16 ± 49.83	244.41 ± 30.13	263.63 ± 15.35	256.91 ± 38.13	244.75 ± 11.91
SV, µL	219.43 ± 27.86	182.83 ± 10.82	210.01 ± 23.71	216.28 ± 21.84	233.88 ± 23.15	206.55 ± 22.86	224.25 ± 28.84	228.74 ± 38.15
CO, mL/min	55.97 ± 6.27	54.97 ± 6.07	63.39 ± 8.80	62.82 ± 11.05	56.93 ± 7.21	54.23 ± 5.61	56.93 ± 2.95	55.90 ± 9.42
LVIDd, mm	7.73 ± 0.53	7.90 ± 0.27	7.96 ± 0.46	8.15 ± 0.52	8.18 ± 0.75	8.73 ± 0.15	8.63 ± 0.26	8.49 ± 0.71
LVIDs, mm	4.66 ± 0.53	* 5.66 ± 0.41	5.37 ± 0.41	5.43 ± 0.52	5.19 ± 0.89	6.36 ± 0.41	6.13 ± 0.20	5.88 ± 0.73
IVSd, mm	1.52 ± 0.05	1.53 ± 0.16	1.59 ± 0.16	1.54 ± 0.23	1.48 ± 0.20	1.36 ± 0.07	1.40 ± 0.16	1.41 ± 0.10
IVSs, mm	2.57 ± 0.11	2.24 ± 0.20	2.55 ± 0.22	2.65 ± 0.27	2.52 ± 0.24	2.19 ± 0.20	2.22 ± 0.21	2.34 ± 0.23
LVPWd, mm	1.66 ± 0.15	1.57 ± 0.12	1.70 ± 0.09	1.98 ± 0.55	1.57 ± 0.17	1.62 ± 0.27	1.57 ± 0.16	1.66 ± 0.12
LVPWs, mm	2.65 ± 0.23	2.36 ± 0.28	2.53 ± 0.07	2.69 ± 0.37	2.42 ± 0.26	2.28 ± 0.26	2.38 ± 0.24	2.58 ± 0.23
E′, mm/s	43.82 ± 5.66	** 32.67 ± 2.14	* 37.82 ± 4.16	36.33 ± 2.95	40.23 ± 1.62	* 32.44 ± 2.62	34.93 ± 3.63	35.94 ± 5.40
E/A ratio	1.66 ± 0.32	2.10 ± 0.40	1.91 ± 0.37	1.75 ± 0.34	1.82 ± 0.28	1.95 ± 0.17	2.04 ± 0.50	2.16 ± 0.64
E/E′ ratio	21.09 ± 0.68	**** 25.74 ± 0.68	**** 22.26 ± 1.50	**** 20.92 ± 1.58	21.14 ± 0.68	*** 25.69 ± 0.54	** 22.50 ± 1.83	*** 21.42 ± 1.29

Values are expressed as the mean ± standard deviation. EF, ejection fraction; FS, fractional shortening; SV, stroke volume; CO, cardiac output; LVIDd, left ventricular internal diameter at diastole; LVIDs, left ventricular internal diameter at systole; IVSd, interventricular septal thickness at diastole; IVSs, interventricular septal thickness at systole; LVPWd, left ventricular posterior wall thickness at diastole; LVPWs, left ventricular posterior wall thickness at systole; E′, early diastolic tissue doppler velocity; E/A, the ratio of the early (E) to late (A) ventricular filling velocities; E/E′, the ratio of the early (E) to early diastolic tissue Doppler velocities. * *p* < 0.05, ** *p* < 0.01, *** *p* < 0.001, **** *p* < 0.0001 by two-way ANOVA test with Tukey’s post hoc correction.

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
