# Peer review of "Sappanone A Prevents Left Ventricular Dysfunction in a Rat Myocardial Ischemia Reperfusion Injury Model"

_ijms, 2020, doi:10.3390/ijms21186935_

Round 1

Reviewer 1 Report

Myocardial infarction is a most serious medical condition of coronary heart disease. Great efforts have been made to identify natural products to lessen myocardial injury. Here, Jo et al. reported that sappanone A could be a promising molecule to be applied for cardiac protection after I/R injury. However, the authors should provide more information to confirm the findings.

  1. Cardiac function and remodeling should be assessed in a long term to confirm the protective effects of sappanone A. Instead of at 4 days, these parameters need to be assessed at 4 weeks after I/R.
  2. It is not reliable and clear to quantify the inflammatory cell infiltration using H&E staining. It is highly recommended to use antibody or special staining for immune cells, eg. neutrophils, macrophages, etc.
  3. The authors claimed that sappanone A protected myocardium via anti-inflammation. They showed that DEGs were enriched in inflammatory pathways using mRNA sequencing. However, this result couldn't fully support this conclusion. The authors should mention the limitation or provide more direct evidence.
  4. TTC/Evan's Blue double staining is highly recommended to assess the infarct size. It helps to normalize the area at risk.
  5. For reliability, the time point of serum sampling (eg. CK-MB, LDH, etc) should be provided after I/R.
  6. Mapping rate, mitochondrial read percentage, total reads etc should be provided to assess the quality of mRNA sequencing.
  7. The key findings from mRNA sequencing should be further validated using qPCR.
  8. From the TTC staining, sappanone A protected myocardium from infarction. Cardiomyocyte death or death related signaling should be assessed.

Author Response

To the editor of International Journal of Molecular Sciences

Please find the enclosed revised manuscript (Manuscript ijms-890925) entitled “Sappanone A prevents left ventricular dysfunction in a rat myocardial ischemia reperfusion injury model”, by Woori Jo, Byung Sun Min, Na-Hye Park, Kyung-Ku Kang, Sehyun Chae*, Eun Sook Ma*, Woo-Chan Son*.

We are pleased to have an opportunity to improve our manuscript according to the reviewers’ suggestions. As the reviewers suggested, we assessed infarct size of heart tissue slices with EB/TTC double staining and more detailed cell type histopathology of inflammatory cells infiltration including lymphocyte and mononuclear cells. Also, we validated differential expression of inflammatory genes found in mRNA seq using RT-PCR. We have revised our manuscript according to the reviewers’ comments. We think that the revised manuscript has been clearer in many aspects due to the reviewers’ comments. Our detailed point-by-point responses to the reviewers’ comments are described below. We appreciate your consideration and look forward to hearing from you.

Sincerely,

Woo-Chan Son, DVM, PhD,

Department of Pathology,

University of Ulsan College of Medicine, Asan Medical Center,

88 Olympic-ro 43-gil, Songpa-gu, Seoul, 138-736, Korea.

Tel. +82-2-3010-4051

Fax. +82-2-3010-8163

Email. wcson32@hanmail.net, wcson@amc.seoul.kr

Reviewer 2 Report

In the present work, Jo and co-workers studied the effect of the homoisoflavone Sappanone A on the development of heart damage after an acute myocardial infarction (MI) in a rat model. Authors described a positive effect of Saponpanone A administration over the 4 days after MI. First, authors evaluated a reduction on the 24h infarct size and serum myocardial damage markers (CK-MB, LDH and AST). Second, they also analyzed histopathological changes (H&E and Masson Trichrome) and cardiac function by echocardiography at day 4. Finally, authors performed an mRNA sequencing analysis to study potential changes in cellular pathways that could help elucidate the mechanism by which Sappanone A exerts its protective effect.

The study is interesting, and results are relevant. However, there are some concerns to clarify:

  • It seems that Sappanone A could act via changing expression of some inflammation related genes. Also, authors described a lower inflammatory response in hearts stained with H&E. Authors must clarify whether the hearts were collected at day 1 or 4 for histopathology. Same for those hearts for the mRAN seq experiment. In this regards, the study will improve if authors stain significant protein related to inflammatory genes (TGFb, TNFa, INFg…) or specific cell markers (such as for macrophages, neutrophils or lymphocytes) since H&E is not such a specific stain for inflammation.
  • Did the authors confirmed any relevant changes found in mRNA seq, either by RT-PCR and protein (western blott or inmuno)? 
  • Did the authors study the effect of Sapponone A on microvascular obstruction or in neutrophil infiltration, or even gene expression on inflammatory cells at early time after reperfussion? This will help to understand whether the effect of Sappanone is driven by a direct effect on inflammatory cells rather than an indirect changes due to change in cardiomyocyte cell survival. In other words, can the authors confirm whether changes in mRNA are directly induced by Sappanone A or by the fact that survival of cardiomyocytes is higher? At the same time, are these changes found in heart samples happening in the cardiomyocytes, or rather induced by a lower inflammation? These facts/speculations/limitations must be included in the discussion.

Minor concerns:

  • Authors must clearly indicate in the methodology if the drug administration begun before ischemia, during the 30min ischemia or after the reperfusion.
  • Described time-point of serum, and for heart collection for mRNA and histopathology in the methodology section.
  • For the mRNA seq, which areas of the LV where selected?
  • The figures will improve if authors represent individual values together with column and SEM. That will help to quickly understand variability, sample dispersion and number.
  • Figure 3 should show different zoom in images to highlight areas of interest. This will help readers to understand the results and the degree of damage.

Author Response

(The authors gave the same response as above.)

Round 2

Reviewer 1 Report

Although some of the recommended assays are still missing, most of the concerns have been clarified using different assays. The current manuscript provides interesting insights into cardiac protection using Sappanone A. It may provide some new cardiac protective strategy in IR.

Author Response

To the editor of International Journal of Molecular Sciences

Please find the enclosed revised manuscript (Manuscript ijms-890925) entitled “Sappanone A prevents left ventricular dysfunction in a rat myocardial ischemia reperfusion injury model”, by Woori Jo, Byung Sun Min, Na-Hye Park, Kyung-Ku Kang, Sehyun Chae*, Eun Sook Ma*, Woo-Chan Son*.

We are pleased to have an opportunity to improve our manuscript according to the reviewers’ suggestions. As the reviewers suggested, we discussed the evaluation of the inflammatory cells infiltration with more detailed cell type by histopathology including lymphocyte and mononuclear cells. Also, we described the detail information of RT-PCR. We hope that the revised manuscript has been clearer in many aspects fulfil the reviewers’ comments. Our detailed point-by-point responses to the reviewers’ comments are described below. We appreciate your consideration and look forward to hearing from you soon.

Sincerely,

Woo-Chan Son, DVM, PhD,

Department of Pathology,

University of Ulsan College of Medicine, Asan Medical Center,

88 Olympic-ro 43-gil, Songpa-gu, Seoul, 138-736, Korea.

Tel. +82-2-3010-4051

Fax. +82-2-3010-8163

Email. wcson32@hanmail.net, wcson@amc.seoul.kr

Reviewer 2 Report

The authors have made good efforts to answer all the concerns raised by the reviewers. However, there is still a bit of room for improvement.

First, the way of assessing the different cell-type infiltration is not the most reliable. Although, I understand there is a limitation of time to perform specific antibody staining, it is not clear the differentiation between lymphocyte and mononuclear cells, since lymphocytes are mononuclear cells itself. This needs a better explanation.

Additionally, I would recommend authors to add on figure 3B a representative zoom cell-image of the different cells they have analyzed. Therefore it will be easier for the reader to understand how authors have evaluated the infiltration.

Moreover, in the text it is not clear whether authors quantified and /or found differences between different cell-type infiltrations. However, it seems Sappanone A influence more on the lymphocyte. Did the authors get consistent results for discussion? 

Last, I appreciate the mRNA analysis by RT-PCR, however it is not clear whether the authors used a qualitative or semi-quantitative method. Authors must fully explain the PCR method (cycles) and how they performed quantification?

Author Response

(The authors gave the same response as above.)
